# Maternal age and the rising incidence of hypertensive disorders of pregnancy: A comprehensive analysis of national claims data from Japan

**Naomi Maeda[1,2], Masayuki Koyama[2,3]\*, Shintaro Takatsuka[4], Keisuke Oyatani[5], Nobuaki Himuro[2], Tasuku Mariya[6], Yoshika Kuno[1,6], Shiro Hinotsu[7], Tsuyoshi Saito[6], Hirofumi Ohnishi[2,3]**

1 Department of Nursing, Sapporo Medical University School of Health Sciences, Sapporo, Japan,
2 Department of Public Health, Sapporo Medical University School of Medicine, Sapporo, Japan,
3 Department of Cardiovascular, Renal and Metabolic Medicine, Sapporo Medical University School of Medicine, Sapporo, Japan, 4 Center of Medical Education, Sapporo Medical University School of Medicine, Sapporo, Japan, 5 Department of Pediatrics, Sapporo Medical University School of Medicine, Sapporo, Japan, 6 Department of Obstetrics and Gynecology, Sapporo Medical University School of Medicine, Sapporo, Japan, 7 Biostatistics and Data Management, Sapporo Medical University School of Medicine, Sapporo, Japan

\* masa3yuki3@sapmed.ac.jp

**Data availability statement:** Data Availability Statement: a) Dataset description: This study analyzed data from the National Database of Health Insurance Claims and Specific Health Checkups of Japan in Hokkaido. The dataset includes medical insurance claims data from patients who visited medical institutions in Hokkaido between April 2010 and March

## Abstract

### Background

Hypertensive disorders of pregnancy (HDP) significantly increase the risk of developing hypertension and cardiovascular disease (CVD) later in life and are a major cause of maternal mortality. However, little is known about the nationwide, long-term, all-inclusive status of HDP.

### Objective

To estimate the incidence of HDP from 2011 to 2019 in Hokkaido, Japan, with a focus on age groups.

### Methods

Using National Database (NDB) insurance medical data, a retrospective analysis was conducted. Due to the absence of direct pregnancy data, birth numbers were used as a surrogate for the number of pregnant women to calculate the incidence of HDP.

### Results

The average incidence rate of HDP over 9 years was 6.37%. The incidence rate was lowest among women aged 25–29 years, at 5.58% (95% confidence interval [CI]: 5.43–5.73), and increased with age, peaking at 10.58% (95% CI: 10.10–11.09) among women over 40 years. Notably, the incidence rate for women under 20 years of age was 6.70% (95% CI: 5.97–7.51), which was higher than that for women in their 20s. A mean annual increase of

2020. These data are under the jurisdiction of the Ministry of Health, Labour and Welfare (MHLW) of Japan. b) Data access permissions: The authors obtained approval from the MHLW to use this dataset (approval number: 0319). The study was also approved by the Ethics Committee of Sapporo Medical University (July 2020, Approval No. 2-1-10). The handling of the data complied with the "Guidelines for the Use of the National Database of Health Insurance Claims and Specific Health Checkups of Japan, Second Edition" [26]. c) Access privileges: The authors did not receive any special privileges in accessing the data. The data were provided in anonymized form, and the Ethics Committee of Sapporo Medical University waived the need for informed consent (July 2020, Approval No. 2-1-10). d) Contact information and access procedures: Access to the raw data is restricted and requires prior approval from the MHLW. Researchers interested in accessing these data should contact: Division for Health Care and Long-term Care Integration Health Insurance Bureau Ministry of Health, Labour and Welfare 1-2-2 Kasumigaseki, Chiyoda-ku, Tokyo, Japan Tel: +81-3-5253-1111 Detailed information about the application process is available at: https://www.mhlw.go.jp/stf/seisakunitsuite/bunya/kenkou_iryou/iryouhoken/reseputo/index.html Important restrictions: - All data must be deleted after study completion as per MHLW guidelines - Release of datasets and intermediate deliverables is not permitted - Only final research outcomes can be published The data that support the findings of this study are available from the MHLW, but restrictions apply to their availability as they were used under license for the current study. For further guidance on the application process, interested researchers may contact either the MHLW directly or the authors with permission from MHLW.

**Funding:** This work was supported by JSPS KAKENHI (grant number JP 21K10938).

**Competing interests:** The authors have declared that no competing interests exist.

0.25% in age-adjusted incidence was observed during this period, which was statistically significant ($R^2 = 0.87$, $p < 0.01$).

## Conclusion

This study reveals that the risk of developing HDP is associated with both older child-bearing and younger pregnancies and follows a J-curve, suggesting that factors other than maternal aging also contribute to the increased incidence of HDP and that further research on risk factors for HDP, which is on the rise worldwide, is urgently needed.

## Introduction

Hypertensive disorders of pregnancy (HDP), which cause hypertension (blood pressure ≥ 140/90 mmHg) during pregnancy, adversely affects mothers and children during the perinatal period and is one of the major causes of maternal death [1–3]. Furthermore, women affected by HDP are at high risk of developing hypertension [4–8] and cardiovascular disease (CVD) [8–13]. For long-term follow-up, it is necessary to properly understand the actual status of HDP patients.

Several studies on the incidence and prevalence of HDP in patients have been reported. The reported prevalence of HDP among pregnant women varies from 5–10% of pregnancies to approximately 5.2–8.2%, but most of these results are based on limited populations at limited time points [14,15]. According to the results of the analysis using data from 1990–2019 for 204 countries around the world, the number of HDP cases is increasing worldwide; in all countries except those with lower sociodemographic indices and human development indices, the age-standardized incidence rate has been reported to be decreasing [16]. However, the age-standardized incidence rate is a value for 100,000 people and is not calculated as a percentage of pregnant women.

One of the risk factors for HDP is advanced maternal age, and the number of patients is estimated to increase in developed countries due to the recent increase in childbearing age [17–19]. Japan is no exception, with the average age of first childbirth consistently increasing over the past decade, reaching 30.9 years in 2021, placing it among the highest average ages in developed countries [20,21].

Taken together, these findings indicate that HDP patients are at increased risk of developing CVD in the future. However, there are not enough reports providing long-term follow-up data on HDP patients at the regional level or detailed incidence rates by age group.

Here, we aim to determine all annual trends in the number of HDP patients by age group and morbidity in the Hokkaido region, the largest island in Japan, using the National Database of Health Insurance Claims and Specific Health Checkups of Japan (NDB).

## Methods

### Data source

This was a retrospective descriptive study using receipt information from the NDB, a Japanese health insurance database. The NDB is the largest database of insured people in Japan; it was created and operated by the Japanese Ministry of Health, Labor and Welfare (MHLW) in 2009 and has been available for research since 2011. Japan has had a universal health insurance system since 1961, and all citizens except welfare recipients are covered by health insurance [22]; as of 2019, welfare recipients accounted for 1.6% of the population and less than 1% of women of reproductive age [23]. In addition, only electronic data are stored in the NDB, and the electronic receipt rate was reported to be 98.3% for hospitals and clinics as of November 2020.

As a result, 25 billion cases of receipt data (2 billion additional cases per year) for 2009–2020, covering more than 95% of the population, are stored in the NDB [24,25].

The data contained in the NDB provide information on each patient's identifier (ID variable), dates of prescriptions and visits, age group, gender, region where procedures were performed, description of these procedures, World Health Organization (WHO) International Classification of Diseases (ICD-10) diagnosis codes, and medical care received. Nevertheless, this database does not include obstetric information such as laboratory data or gestational age.

We obtained approval from the MHLW to obtain medical insurance data for patients who visited medical institutions in Hokkaido between April 2010 and March 2020 (approval number: 0319). Additionally, the study was approved by the Ethics Committee of Sapporo Medical University (July 2020, Approval No. 2-1-10). The need for informed consent was waived because all data were provided to the authors in anonymized form, so the need for informed consent was waived by The Ethics Committee of Sapporo Medical University. The handling of the data complied with the guidelines [26] indicated by the MHLW. The date of access to NDB data for the purpose of this study is 13 August 2021.

## Data preparation

Two main IDs (ID1 and ID2) are used in the NDB as variables that link multiple receipts of the same patient. ID1 is based on the insurer number, insurance card number, date of birth, and gender, while ID2 is based on the name, date of birth, and gender information, each of which is encrypted and attached. In Japan, an insured person may switch insurers due to a change in the workplace or a change in family name upon marriage. This implies that there will be a certain number of individuals with the same condition but different IDs during the follow-up period. Therefore, it has been noted that one of the problems with analysis by the NDB is counting ID1 and ID2, as they exceed the actual number of patients. Attempts to more accurately count patients on the NDB have been reported, including studies that created ID0 and virtual patient identifier to count patients [27,28], a study to count the number of patients by integrating monthly bills issued by each institution for each individual when ID1 and ID2 are the same [29], and a study that counted ID1s, treating different ID1s as identical if they shared the same ID2, etc [30]. However, in all these methods, true information cannot be obtained because it is impossible to match with actual patient data, and there are limits to the pursuit of accuracy. In this study, IDT was created by concatenating ID1 and ID2 as much as possible. Even if the ID1 is different, if the ID2 is the same, it is considered to be the same patient, and even if the ID2 is different, if the ID1 is the same, it is considered to be the same patient (or the ID is reassigned so that it can be considered to be the same). In practice, the following procedure is used.

STEP 0. All data rows have the ID1, ID2, and IDT columns. Initially, IDT is empty.
STEP 1. Assign the same IDT to the same ID1.
STEP 2. Make sure that the same IDT is used for the same ID2.
STEP 3. Make sure that the same IDT is used for the same ID1.
STEP 4. Repeat steps 2 and 3 until the number of updates reaches zero.

The IDT is the theoretical lower limit of the number of persons to be collated, and the upper limit of the number of persons to be collated (the number of ID1 or ID2 counted) is also shown, thus allowing a range of accuracy to be pursued. We attempted to pursue accuracy with the IDT.

## Definition of HDP patients

In Japan, HDP is classified into four forms: gestational hypertension (GH), preeclampsia (PE), superimposed preeclampsia (SPE), and chronic hypertension (CH) [31,32]. This complies

with the international definitional classification and was revised in 2018. In this study, the researchers reviewed and determined 15 Japanese standardized disease codes. These diseases do not include related diseases, such as eclampsia or HELLP syndrome, and are classified into four disease types: GH, PE, SPE, and CH. The researchers decided to treat patients with HDP if they had at least one of these 15 disease codes, and the "medical treatment start date" information was included in the receipts for the same month. The injury/disease name, injury/disease code, and corresponding International Classification of Diseases, 10th revision (ICD-10 code) are shown in Table 1 below.

Next, to count HDP patients, we defined one gestational period on receipt. Because this study included data collected over 9 years, it included one woman who had multiple pregnancies and developed HDP. To count HDP due to another pregnancy as a separate patient, it is necessary to know the termination of pregnancy. However, in Japan, normal pregnancy and vaginal delivery are considered healthy life events and are not covered by insurance. Therefore, it is difficult to ascertain the termination of pregnancy from the obtained receipt information because no receipt is generated when a normal delivery occurs among those with an HDP injury or disease.

**Table 1. Number of Patients by HDP Disease Code, 2011–2019.**

| Disease Code | ICD-10 Description | ICD-10 Code | 2011 | 2012 | 2013 | 2014 | 2015 | 2016 | 2017 | 2018 | 2019 |
|---|---|---|---|---|---|---|---|---|---|---|---|---|
| 8845462 | Pre-existing essential hypertension complicating pregnancy, childbirth and the puerperium | O100 | – | 12 | 19 | 14 | 19 | 22 | 27 | 21 | 25 |
| 8845458 | Pre-existing hypertensive heart disease complicating pregnancy, childbirth and the puerperium | O101 | – | – | – | – | – | – | – | – | – |
| 8845460 | Pre-existing hypertensive renal disease complicating pregnancy, childbirth and the puerperium | O102 | – | – | – | – | – | – | – | – | – |
| 8845459 | Pre-existing hypertensive heart and renal disease complicating pregnancy, childbirth and the puerperium | O103 | – | – | – | – | – | – | – | – | – |
| 8845461 | Pre-existing secondary hypertension complicating pregnancy, childbirth and the puerperium | O104 | – | – | – | – | – | – | – | – | – |
| 8842687 | Pre-eclampsia superimposed on chronic hypertension | O11 | 12 | 16 | 13 | 16 | 19 | 15 | 27 | 21 | 17 |
| 6429003 | Gestational hypertension | O13 | 406 | 402 | 363 | 389 | 469 | 382 | 374 | 416 | 382 |
| 8842709 | Mild to moderate pre-eclampsia | O140 | 54 | 76 | 104 | 106 | 104 | 102 | 128 | 122 | 131 |
| 8848335 | Mild to moderate pre-eclampsia | O140 | – | – | – | – | – | – | – | 17 | 24 |
| 8842765 | Severe pre-eclampsia | O141 | 122 | 150 | 204 | 200 | 245 | 267 | 279 | 257 | 355 |
| 8848357 | Severe pre-eclampsia | O141 | – | – | – | – | – | 21 | 30 | 38 | 54 |
| 8842791 | Pre-eclampsia, unspecified | O149 | – | – | – | – | – | – | – | – | – |
| 8842804 | Pre-eclampsia, unspecified | O149 | – | – | – | – | – | – | 12 | – | 11 |
| 8842827 | Pre-eclampsia, unspecified | O149 | 1935 | 1857 | 2080 | 1943 | 1943 | 2008 | 2080 | 2059 | 1682 |
| 8842828 | Pre-eclampsia, unspecified | O149 | 143 | 82 | 103 | 139 | 207 | 224 | 219 | 234 | 246 |

This table shows the number of patients with HDP with each diagnosis categorized by domestic diagnosis codes for healthcare claims in Japan. The Japanese standardized disease codes are part of a comprehensive classification system used in Japan to identify and record diseases and conditions. These codes are linked to the International Classification of Diseases, 10th Revision (ICD-10), but include additional details to accommodate Japan's healthcare system and billing practices. While some ICD-10 codes directly map to a single Japanese standardized disease code, others may correspond to multiple codes depending on the context and specific details recorded in medical receipts. For instance, codes for gestational hypertension and preeclampsia may vary based on the level of detail provided in the diagnosis or treatment description. To ensure consistency, this study reviewed and standardized 15 Japanese disease codes explicitly associated with hypertensive disorders in pregnancy (HDP). The mapping process involved careful evaluation of the disease descriptions and their correspondence to ICD-10 codes. This approach ensures that the selected codes accurately represent HDP cases within the constraints of the NDB. According to MHLW's rules for publication of NDB data, we did not show the number of cases in categories with less than 10 (indicated by "-" in the table). The sum of patients is not equal to the total number of patients because some patients are given two or more diagnoses.

HDP; Hypertensive Disorders of Pregnancy, MHLW; the Japanese Ministry of Health, Labor and Welfare, NDB; National Database.

Therefore, in this study, we defined the duration of pregnancy based on the "medical care start date" information recorded on the receipts. The information on the 'date of start of treatment' is given not only once per pregnancy but also when the medical institution changes or the name of the injury or disease differs. Hence, there is a threshold where pregnancies with the same ID and within X months of the earliest treatment start date are counted as the same, but pregnancies over X + 1 month are counted as different pregnancies. To determine this X, the number of IDs was calculated and compared at 6, 7, 8, 9, 10, 11, 12, and 13 months after the start of treatment as separate pregnancies, and the difference in the number of IDs at approximately 10 months after counting as separate pregnancies was minimal (Fig 1). Based on the above, in this study, pregnancies within 9 months of the earliest clinical start date were defined as the same, while pregnancies of 10 months or more were considered separate pregnancies.

The number of IDs calculated by the above method was defined as the "number of HDP patients" in this study.

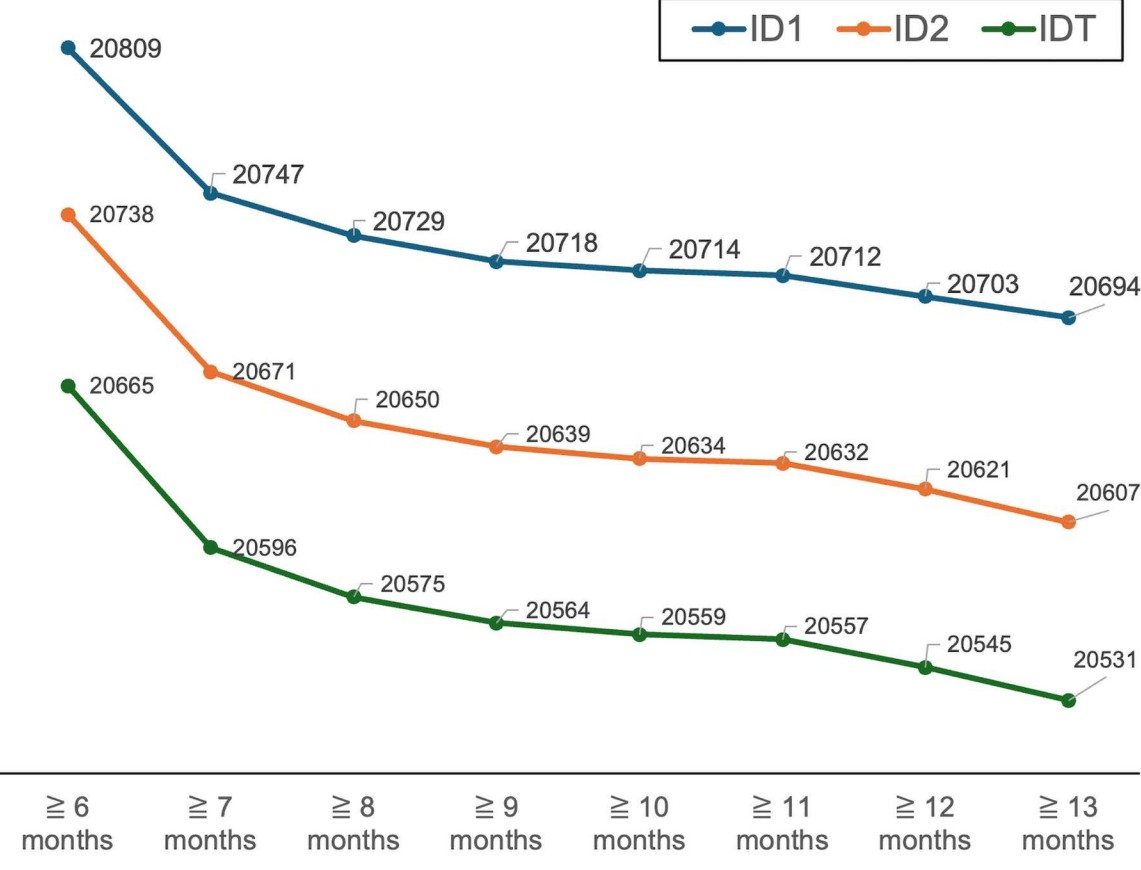

**Fig 1. ID Counts for Periods Considered as Separate Pregnancies.** ID counts were calculated and compared for separate pregnancies counted after 6, 7, 8, 9, 10, 11, 12, and 13 months from the start of medical care. When pregnancies were treated as separate starting at a 10-month interval, the difference in ID counts was the smallest between adjacent periods.

### Data analysis

The number of HDP patients by year and the number of patients and morbidity by age group at the start of practice from January 2011 to December 2019, before the COVID-19 epidemic, were tabulated to obtain confidence intervals and age-adjusted morbidity rates. Linear regression analysis was used for annual incidence rates, the Kruskal-Wallis test was used to test for differences in incidence rates across all age intervals, and the z-test for differences in proportions was used to analyze differences in incidence rates between specific age groups. The statistical analysis performed in this study used Python 3.12.0. The significance level was set at 5% in the statistical tests, and the Holm method correction was applied for multiple comparisons of incidence rates among age groups.

The age information provided by the NDB is based on 5-year age categories. Nevertheless, due to the MHLW's rule that numbers less than 10 cannot be published, those aged ≤14 and ≥45, including categories with a value of less than 10, were grouped into the ≤ 19 and ≥ 40 age groups, respectively. Thus, the six age categories of ≤ 19, 20–24, 25–29, 30–34, 35–39, and ≥ 40 years were used in this study.

The incidence of HDP is determined by the number of HDP cases among pregnant women. However, since the total number of pregnant women on the NDB is unknown because the receipts do not include pregnant women with a normal pregnancy, we utilized publicly available statistical data. In Japan, the "number of pregnancy notifications" and the "number of deliveries" after 12 weeks of pregnancy are compiled, but these publicly available data do not include maternal age information. Therefore, in this study, the number of births that are close to these data and have maternal age information is used as the number of pregnant women, and the incidence rate of HDP is defined as the number of HDP cases per year as a percentage of the number of births per year. For the calculation of age-adjusted incidence rates, the average total number of births in Japan from 2011 to 2019 and the average number of births by mother's six age categories were used as the reference number of births (S1 Table). The average number was used because the change in the number of live births during this period significantly decreased, decreasing by approximately 17%.

### Results

The average number of HDP patients in Hokkaido in each year from 2011–2019 was determined by IDT, ID1, and ID2, with a minimum of 20,559 for IDT and a maximum of 20,714 for ID1 over the 9 years (Fig 2). The incidence rates were 6.37% for the smallest number, IDT, and 6.42% for the largest number, ID1 (Table 2). During this period, the incidence of HDP increased by 0.27% (slope = 0.003, p < 0.01), with a coefficient of determination of approximately 0.89. The number of patients in each of the 15 injury and disease categories in each year is shown in Table 1.

The age composition of the nine-year-old HDP patients was determined by age groups ≤ 19, 20–24, 25–29, 30–34, 35–39, and ≥ 40 years. The results showed that the percentage of the ≥ 40 age group increased from 5.7% in 2011 to 8.8% in 2019, and the percentage of the 25–29 age group decreased from 27.0% to 22.8%, while there were no significant changes in the other groups. (Fig 3). The mean incidence rates and confidence intervals for each age group over this approximate 10-year period are shown in Fig 4. The incidence rate was lowest in the 25–29 age group at 5.58% (95% confidence interval [CI]: 5.43–5.73) and increased with age, with a rate of 10.58% (95% CI: 10.10–11.09) in the 40 + age group. On the other hand, the rate was 6.70% (95% CI: 5.97–7.51) in the group aged 19 years or younger, which was higher than that in the group aged 20 years or older. To evaluate differences in incidence rates across all age groups, a Kruskal-Wallis test was performed, and significant differences were found

NDB in Hokkaido, FY 2010-2019
Medical receipts; 393,451,749; DPC receipts; 5,308,629

**Step 1.** Create a merged ID, denoted as IDT, by concatenating ID1 and ID2 whenever possible.

ID1 (N) = 15,066,584 (Women; 8,183,645 [54.3%])
ID2 (N) = 11,045,827 (Women; 5,992,166 [54.2%])
**IDT (N) = 6,869,645 (Women; 3,600,091 [54.2%])**

**Step 2.** From Medical and DPC receipts, specifically for the period of January 2011 to December 2019, 25,494 receipts were extracted with HDP-related diagnoses and treatment initiation information in the same month.

**Step 3.** Excluded
10 receipts aged under 10 and over 60 (Details undisclosed for those under 10)

**Step 4.** From 25,484 receipts with HDP-related diagnoses and treatment initiation information, receipts with the same ID and a gap of 10 months or more from treatment initiation were treated as separate pregnancies (separate patients), and the number of unique IDs was counted.

**Patients with HDP in Hokkaido from 2011-2019**
ID1 (N) = 20,714
ID2 (N) = 20,634
**IDT (N) = 20,559**

**Fig 2. Study flowchart.** In this analysis, 'receipt serial number types' represent the total number of receipts collected, while 'ID types' indicate the count of unique patient IDs. 'Medical receipts' include records for both outpatient and inpatient services. In contrast, 'DPC receipts are exclusively associated with inpatient services. NDB; National Database, DPC; Diagnosis Procedure Combination, HDP; Hypertensive Disorders of Pregnancy.

**Table 2. Number of HDP Patients in Hokkaido, Japan from 2011–2019.**

| Number of births in Hokkaido | | 2011 | 2012 | 2013 | 2014 | 2015 | 2016 | 2017 | 2018 | 2019 | Total |
|---|---|---|---|---|---|---|---|---|---|---|---|
| | | 39,292 | 38,686 | 38,190 | 37,058 | 36,696 | 35,129 | 34,058 | 32,642 | 31,020 | 322,771 |
| Number of patients with HDP Incidence rate (%) | IDT | **2,156** | **2,075** | **2,268** | **2,214** | **2,395** | **2,363** | **2,446** | **2,471** | **2,171** | **20,559** |
| | (%) | **(5.49)** | **(5.36)** | **(5.94)** | **(5.97)** | **(6.53)** | **(6.73)** | **(7.18)** | **(7.57)** | **(7.00)** | **(6.37)** |
| | ID1 | 2,175 | 2,095 | 2,293 | 2,232 | 2,409 | 2,378 | 2,457 | 2,488 | 2,187 | 20,714 |
| | (%) | (5.54) | (5.42) | (6.00) | (6.02) | (6.56) | (6.77) | (7.21) | (7.62) | (7.05) | (6.42) |
| | ID2 | 2,165 | 2,078 | 2,276 | 2,224 | 2,408 | 2,376 | 2,455 | 2,475 | 2,177 | 20,634 |
| | (%) | (5.51) | (5.37) | (5.96) | (6.00) | (6.56) | (6.76) | (7.21) | (7.58) | (7.02) | (6.39) |

The incidence rates in the table are calculated by dividing the number of patients with HDP by the total number of births in Hokkaido for each year.

HDP; Hypertensive Disorders of Pregnancy.

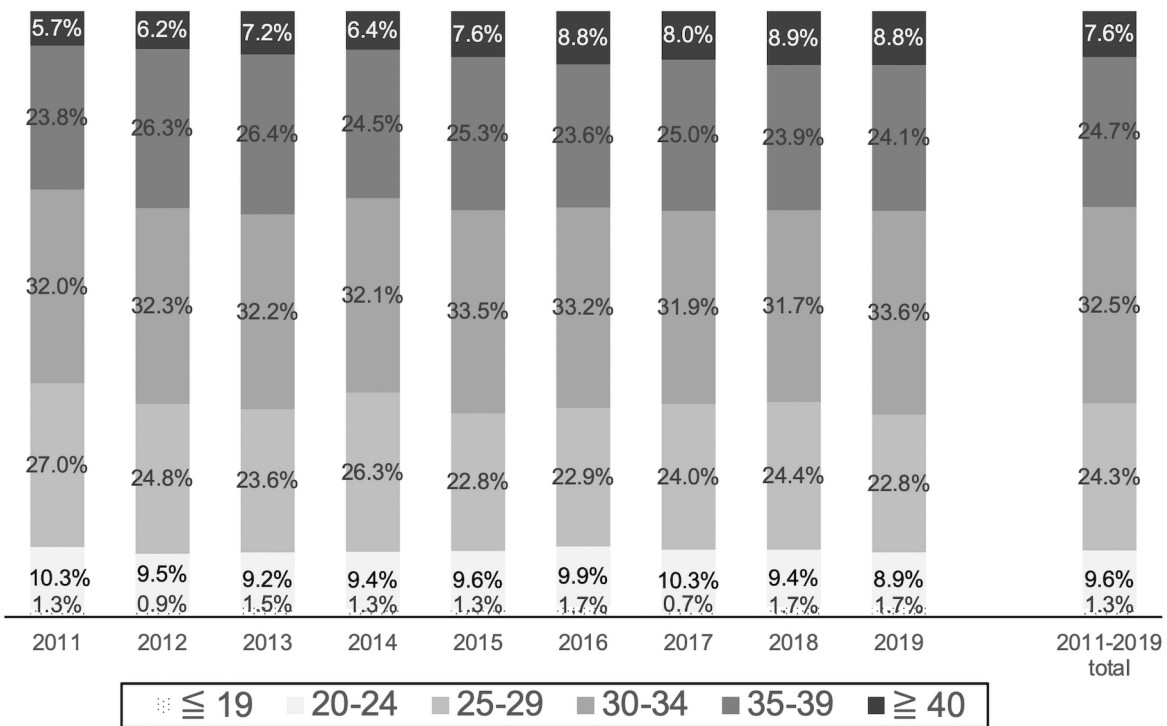

**Fig 3. Age composition of HDP patients, 2011–2019.** The chart shows the age distribution of HDP patients in Hokkaido from 2011 to 2019, with individual years and the average for the entire period. HDP; Hypertensive Disorders of Pregnancy.

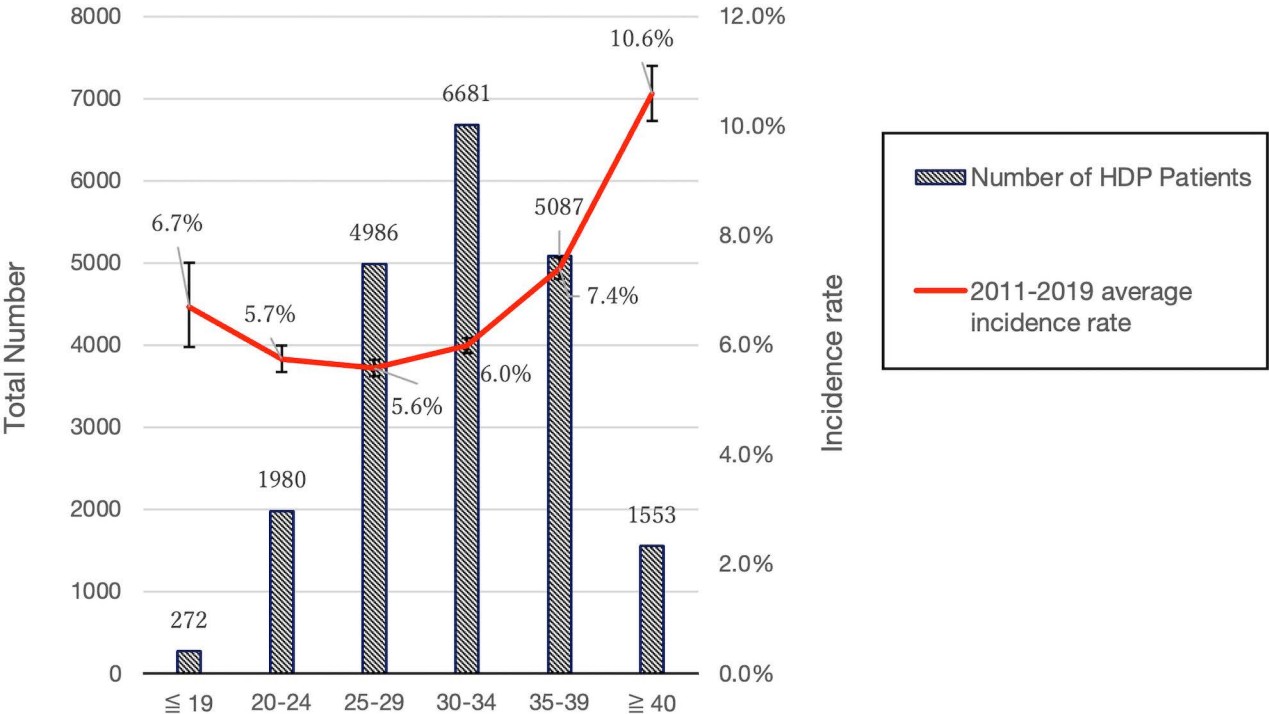

**Fig 4. Number of HDP Patients and Average Incidence Rate.** This figure presents the number of HDP patients alongside the average incidence rates for 2011–2019, with 95% confidence intervals calculated using the Wilson score method for age-specific rates. HDP; Hypertensive Disorders of Pregnancy.

(Kruskal-Wallis test quantity 59.0, p < 0.01). In addition, z-tests for differences in proportions were performed between the 25–29 age group and each of the other five age groups, and multiple comparison correction was applied using the Holm method. The results showed that there was a significant difference in proportions between ≤ 19 years (z = −3.016, p = 0.0025, adjusted p = 0.0051), 20–24 years (z = −1.118, p = 0.26, adjusted p = 0.26), 30–34 years (z = −3.893, p < 0.0001, adjusted p = 0.00030), 35–39 years (z = −14.715, p < 0.0001, adjusted p < 0.0001), and ≥ 40 years (z = −23.139, p < 0.0001, adjusted p < 0.0001), and there were statistically significant differences between the ≤ 19, 30–34, 35–39, and ≥ 40 age groups and the 25–29 age group. The data on the number of patients and incidence rates by age group for each year are presented in Table 3.

The age-adjusted HDP incidence rates for this period were calculated, and the linear regression analysis results are shown in Fig 5. The age-adjusted incidence rates were 5.58% in 2011 and 6.94% in 2019, indicating an increase of 0.25% per year.

## Discussion

Using data from the National Database, which contains insurance practice data for most of the Japanese population, we were able to show age-specific patient numbers and incidence rates for HDP in the Hokkaido region of Japan from 2011–2019. To our knowledge, this is the first report of a comprehensive HDP study using this scale. Hokkaido is the largest island in Japan, accounting for 22% of the total land area, and is surrounded by the sea, making overland travel possible only by rail. Therefore, except for relocation and homecoming deliveries (the Japanese custom of returning to a woman's parents' home during late pregnancy or childbirth to receive support from family members), there is almost no outflow of people into and out of the prefecture to receive medical care. Therefore, the annual trends in the number of HDP patients in this region over a given period may approximate the trends in the number of HDP patients in Japan, a developed country.

**Table 3. Number of Patients with HDP by Age Group, 2011–2019.**

| Age groups (%) | 2011 | 2012 | 2013 | 2014 | 2015 | 2016 | 2017 | 2018 | 2019 | 2011–2019 total |
|---|---|---|---|---|---|---|---|---|---|---|
| ≦ 19 | 27 | 18 | 34 | 28 | 30 | 39 | 18 | 41 | 37 | 272 |
| | (5.11%) | (3.39%) | (6.51%) | (5.53%) | (6.61%) | (8.65%) | (5.16%) | (10.70%) | (10.95%) | (6.70%) |
| 20–24 | 221 | 198 | 208 | 209 | 230 | 235 | 253 | 233 | 193 | 1,980 |
| | (4.66%) | (4.66%) | (5.02%) | (5.42%) | (6.08%) | (6.43%) | (7.23%) | (6.87%) | (6.12%) | (5.74%) |
| 25–29 | 583 | 514 | 535 | 582 | 546 | 540 | 588 | 603 | 495 | 4,986 |
| | (5.02%) | (4.56%) | (4.86%) | (5.68%) | (5.51%) | (5.72%) | (6.51%) | (7.01%) | (6.03%) | (5.58%) |
| 30–34 | 689 | 671 | 730 | 711 | 802 | 784 | 781 | 783 | 730 | 6,681 |
| | (5.15%) | (5.08%) | (5.66%) | (5.55%) | (6.26%) | (6.40%) | (6.49%) | (6.90%) | (6.76%) | (5.99%) |
| 35–39 | 513 | 545 | 598 | 543 | 605 | 557 | 611 | 591 | 524 | 5,087 |
| | (6.60%) | (6.81%) | (7.45%) | (6.86%) | (7.52%) | (7.41%) | (8.29%) | (8.24%) | (7.63%) | (7.41%) |
| ≧ 40 | 123 | 129 | 163 | 141 | 182 | 208 | 195 | 220 | 192 | 1,553 |
| | (9.82%) | (8.96%) | (10.16%) | (8.17%) | (10.71%) | (11.49%) | (11.06%) | (12.67%) | (11.69%) | (10.58%) |

The percentages in the table represent the incidence rates for each age group of mothers, calculated as the number of HDP patients divided by the number of births within the same age category.

HDP; Hypertensive Disorders of Pregnancy.

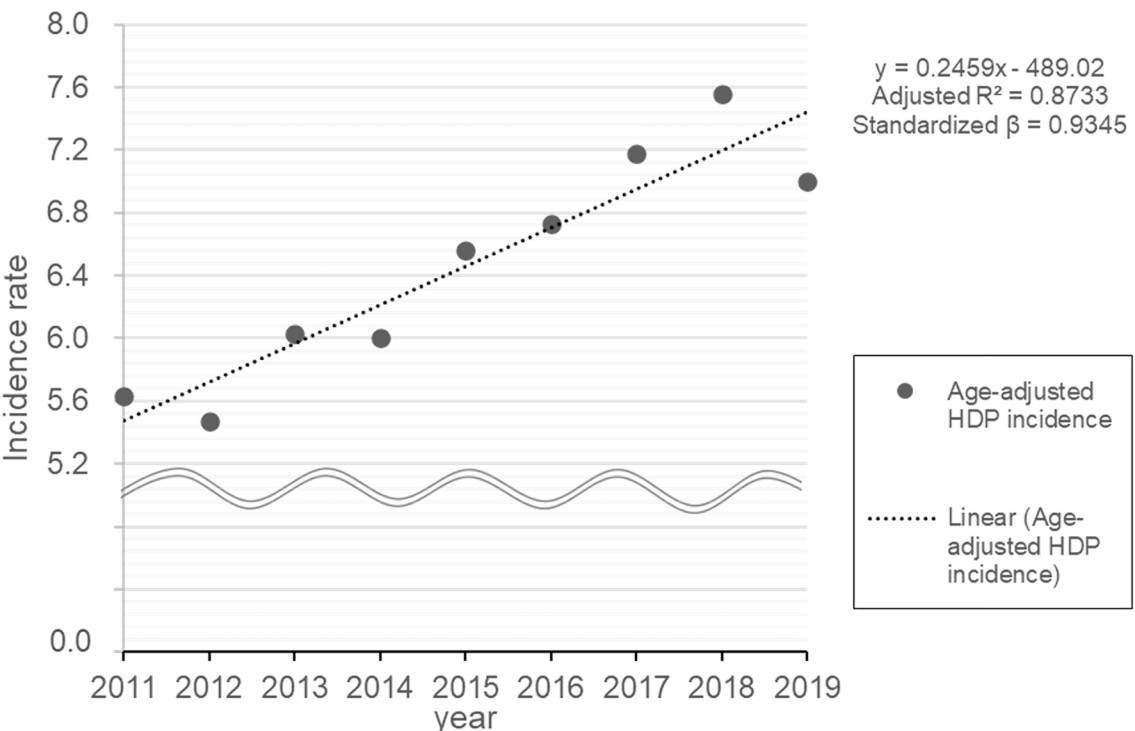

**Fig 5. Age-adjusted HDP incidence rate.** The age-adjusted incidence rates were calculated using the average birth data for six age groups ($\leq$ 19, 20–24, 25–29, 30–34, 35–39, $\geq$ 40) in Japan from 2011 to 2019. HDP; Hypertensive Disorders of Pregnancy.

In this study, the number of HDP cases initiated during a year divided by the number of births during the year was treated as the HDP incidence rate, and the resulting 9-year average incidence rate was 6.37% (Table 2). According to a database created by the Japan Society of Obstetrics and Gynecology, which registers approximately 20% of births in the population of higher tertiary hospitals, there were 108,562 cases of HDP over the same 9-year period, representing 5.79% of all births after 22 weeks and 5.82% of live births [33]. The prevalence of HDP in the Hokkaido Birth Cohort Study reported by Poudel K *et al.* was 1.7%, which was lower than that in this study (6.8%) [34]. The previous study had a different disease background than the present study, with a frequency of 20% in mothers over 35 years of age, which was consistent with results based on a much older database from 2002 to 2013 [35]. Considering the retrospective trend of a gradual increase in HDP from year to year, as shown in Fig 5, as well as the margin of error due to differences in sample size, these differences are considered acceptable. For other countries, the prevalence of HDP in Ireland in 2016 was 5.9%, and a cohort study of births in the United States from 1976–1982 reported an HDP incidence of 7.3% [33,35,36]. Many reports have also fluctuated in the approximate range of 5–10%, with a review by Umesawa *et al.* reporting a rate of approximately 5.2–8.2% [14,15]. Although the diagnostic criteria and classifications of HDP have varied slightly over time and from country to country, the methods used to calculate incidence rates in various studies have not been completely consistent. Nevertheless, even considering this, the results of this study are within the range of those of previous reports and support the validity of the present data.

The findings of this study highlight significant age differences in the incidence of HDP in Hokkaido. In this study, the mean incidence rate by age group was 5.58% (95% CI: 5.43–5.73), with the lowest in the 25–29 age group, 10.58% (95% CI: 10.10–11.09) in the over

40 age group, and even 6.70% (95% CI: 5.97–7.51) in the under 19 age group, which was significantly greater than that of the 25–29 age group (Fig 4). A study analyzing HDP incidence rates in 204 countries around the world reported that the lowest estimated incidence rate was in the 25–29 age group, with higher rates in younger and older age groups [16], and similar results were obtained in this highly universal study limited to Japan, a developed country. The increased risk of HDP in women over 40 years of age, and even in women under 20 years of age, suggests the existence of a multifaceted risk profile that goes beyond simple demographic categories. This "J-curve" phenomenon in morbidity contradicts conventional risk perceptions and indicates that factors other than maternal aging may contribute to the development of HDP. It has already been reported that maternal advanced age is a risk factor for hypertension during pregnancy [15,17–19,37], and study reports on teenage morbidity, including those from developing countries with high rates of young childbearing, have reported a greater risk of HDP and eclampsia compared to those in their 20s [16,38,39]. On the other hand, there are few reports on HDP in teenagers, especially in developed countries. In Japan, young pregnancies account for less than 1% of all pregnancies, but there are many sociologically high-risk pregnancies [40,41], especially smoking, which is known to increase the risk of developing HDP [42,43]; however, the smoking rate among pregnant teenage women is high [44,45], and Hokkaido is the region with the highest female smoking rate in Japan [46]. Another reason for the higher incidence among young women may be the high smoking rate and inadequate education for young pregnant women. The pathogenesis of HDP is known to involve immune tolerance deficiency [47], uterine spiral artery remodeling defects [48], and decreased blood flow and hypoxia in the placental region due to increased production of the anti-angiogenic factors soluble fms-like tyrosine kinase 1 (sFlt-1) [48] and soluble endoglin (sEng) [49–51]. Indeed, significant angiogenic factor imbalances have been reported in pregnant women who develop HDP. While previous reports indicate that the production of sFlt-1, a protein associated with various health conditions, is affected by factors such as advanced glycation end products (AGEs) in elderly individuals [52], the relationship between young pregnancy and sFlt-1 remains unclear. Taken together, it can be inferred that while new factors that may explain the J-curve phenomenon need to be scrutinized, risky health behaviors are related in no small way to the development of HDP and should receive special attention.

According to the current analysis of NDB data for the 9 years before the COVID-19 epidemic, the incidence of HDP increased even after age adjustment (Fig 5). Although the incidence increased to 8.8% in 2019, the increase in incidence even after age adjustment suggests that the main reason is a factor other than gestational age. A retrospective cohort study using the U.S. birth database reported that the prevalence of HDP increased from 4.5% in 2014 to 6.0% in 2018, a trend similar to our results [53]. A study examining the prevalence of HDP in urban and rural areas of the United States also reported a doubling of prevalence in both regions between 2007 and 2019, the study period [54]. One of the factors contributing to the annual increase in incidence in Japan is the increase in assisted reproductive technology (ART), which has been reported to have a significantly greater incidence of HDP than spontaneous conception [55,56]. In 2011, ART births accounted for 3.1% of all births in Japan, but this percentage has increased to 7.0% over the past decades [57]. It should be emphasized that in the future, we can expect to see an increasing number of pregnancies on ART and an increasing number of HDP patients in developed countries.

In recent years, appropriate management of HDP has reduced severe effects, such as maternal mortality. Nevertheless, it has been reported that a 10 mmHg increase in mean diastolic blood pressure during pregnancy increases the risk of hypertension later in life by 1.7-fold, making HDP a subject that needs to be followed throughout pregnancy and later in life,

regardless of its severity [58]. It is necessary to pay attention to the increasing trend of HDP incidence in Japan and worldwide and to link this to public health measures in each country, especially in terms of maternal and child health.

## Strengths and limitations

This study addresses a significant gap in existing research by evaluating the incidence of HDP among younger populations, a group that has been largely overlooked in previous studies. Existing research has predominantly focused on older pregnant women or general population-based estimates, often lacking detailed age-stratified analyses. For instance, international studies such as the Global Burden of Disease project have provided valuable insights into overall prevalence trends but have not adequately captured rare conditions like HDP in adolescents. In contrast, NDB allows for comprehensive analyses across a broad range of age groups, including younger individuals, enabling the detailed identification of rare cases and age-specific trends. Leveraging this capability, the present study tracked HDP incidence rates in younger populations over nine years, addressing a critical gap in public health research.

Moreover, one of the strengths of the NDB lies in its foundation on Japan's universal health insurance system, which covers nearly the entire population. This minimizes selection bias while ensuring robust statistical power. The NDB has been widely recognized as a valuable resource for understanding healthcare utilization trends and disease incidence [28–30,59,60] By utilizing real-world clinical data, it provides critical insights into disease management and patient behavior, complementing findings from prospective cohort studies. This study harnessed these features of the NDB to calculate HDP incidence rates in the Hokkaido region, offering a unique perspective on the epidemiology of HDP.

Another notable strength of this study is its ability to accurately track the annual trends of HDP over a long period, from 2011 to 2019, prior to the onset of the COVID-19 pandemic. Importantly, the study also includes data covering pregnancies among women aged 19 years or younger, an age group that has been challenging to analyze in prior prospective cohort studies conducted in developed countries. This comprehensive inclusion of younger age groups highlights the distinct contribution of this study to understanding the epidemiology of HDP across diverse demographic categories.

However, our study has several limitations. First, the study did not include specific tests or prescriptions and treated patients with HDP, such as those who had any of the 15 injury or disease names and medical care initiation information in the same month. Therefore, it was impossible to classify severity or disease type, and the number of patients included in the study ranged from mild patients who did not require treatment to severe patients. Second, the NDB data used in this study did not include detailed data such as patient symptoms, blood pressure readings, or laboratory results. Therefore, patients with HDP-related injuries and illnesses were considered HDP patients. Nevertheless, it was not possible to confirm whether the injury or illness name correctly reflected the diagnosis, and no validation study was conducted. Third, since normal delivery was not covered by insurance, the date of delivery could not be determined from the receipt information, and the duration of pregnancy was uniformly defined as the same pregnancy from the start date of medical care to nine months. This is illustrated in Fig 1 for clarity, but the number of HDP patients may differ depending on the setting of this cutoff value. Fourth, the lack of data on patient background makes it difficult to estimate factors associated with increased morbidity. Despite these limitations, this study is significant because it is the world's first long-term all-inclusive survey of a specific geographic region, and it allowed us to determine the incidence rates by age over nine years. Finally, the data used in this study are electronic data obtained from insured patients and do not include

data on welfare recipients who do not use insurance or on receipts that are not electronic. Since the percentage of welfare recipients [61] and the rate of electronic receipt [24,62] did not change significantly between 2011 and 2019, we believe that the impact of these changes can be treated as negligible. In addition, the definitional classification of HDP changed in Japan in 2018; however, we infer that this effect is also not significant because we treated all the data based on that classification.

## Conclusion

Analysis using the Japanese Health Insurance Database revealed that the average incidence of HDP in 2011 and 2019 was 6.37%, with the lowest incidence in the 25–29 age group and a "J- curve" with a greater incidence in the younger and older age groups. Age-adjusted morbidity increased slowly over this period, suggesting an association with factors other than older maternal age. Further approaches to identify factors, as well as interventions for existing risk factors, are needed for this disease, which is expected to be on the rise worldwide.

## Supporting information

**S1 Table. Number of Pregnancy Notifications, Deliveries, and Age-specific Births in Hokkaido.** This table presents the number of pregnancy notifications, delivery cases, births, and age-specific birth counts of mothers in Hokkaido, based on publicly available data from 2011 to 2019. It includes annual figures for each year as well as the average over the period. The rightmost column displays the national averages for Japan from 2011 to 2019.
(TIFF)

## Author contributions

**Conceptualization:** Naomi Maeda, Masayuki Koyama, Keisuke Oyatani, Hirofumi Ohnishi.

**Data curation:** Naomi Maeda, Masayuki Koyama, Shintaro Takatsuka, Keisuke Oyatani.

**Formal analysis:** Naomi Maeda, Masayuki Koyama, Shintaro Takatsuka, Keisuke Oyatani.

**Funding acquisition:** Naomi Maeda, Masayuki Koyama, Hirofumi Ohnishi.

**Investigation:** Naomi Maeda, Masayuki Koyama.

**Methodology:** Naomi Maeda, Masayuki Koyama, Shintaro Takatsuka, Nobuaki Himuro, Tasuku Mariya, Yoshika Kuno, Shiro Hinotsu, Hirofumi Ohnishi.

**Project administration:** Naomi Maeda, Masayuki Koyama, Hirofumi Ohnishi.

**Resources:** Naomi Maeda, Masayuki Koyama, Shintaro Takatsuka, Nobuaki Himuro, Shiro Hinotsu, Tsuyoshi Saito, Hirofumi Ohnishi.

**Software:** Naomi Maeda, Masayuki Koyama, Shintaro Takatsuka, Keisuke Oyatani, Nobuaki Himuro.

**Supervision:** Masayuki Koyama, Shintaro Takatsuka, Nobuaki Himuro, Tasuku Mariya, Yoshika Kuno, Tsuyoshi Saito, Hirofumi Ohnishi.

**Validation:** Naomi Maeda, Masayuki Koyama, Shintaro Takatsuka, Shiro Hinotsu, Tsuyoshi Saito.

**Visualization:** Naomi Maeda, Masayuki Koyama, Keisuke Oyatani.

**Writing – original draft:** Naomi Maeda.

**Writing – review & editing:** Masayuki Koyama, Keisuke Oyatani, Nobuaki Himuro, Tasuku Mariya, Yoshika Kuno, Shiro Hinotsu, Tsuyoshi Saito, Hirofumi Ohnishi.

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
