## [Decision Letter · Decision Letter 0]

6 Jan 2025

PONE-D-24-52545
Maternal Age and the Rising Incidence of Hypertensive Disorders of Pregnancy: A Comprehensive Analysis of National Claims Data from Japan.
PLOS ONE

Dear Dr. Koyama,

Thank you for submitting your manuscript to PLOS ONE. After careful consideration, we feel that it has merit but does not fully meet PLOS ONE’s publication criteria as it currently stands. Therefore, we invite you to submit a revised version of the manuscript that addresses the points raised during the review process.

Thank you for submitting your manuscript to plos one. Please note that the following corrections are compulsory together with a detailed response to all the Reviewers recommendations (kindly ensure that that thr response shows what changes were made to the manuscript if any, appropriate justification to each comment must be given.)

1. The statistical analysis applied must be substantiated by supplying the data distribution trends and appropriate justification for the statistical tests applied. 

2. The discussion lacks critical analysis and this must be corrected so that the novelty of the research incomparison to work already done can be justified.  There are a number of studies done on the prevalence of HDPs globally. This can be used in the critical discussion, and how does your manuscript address the gaps that previous research has over looked.  In addition discuss how claims data can be beneficial to scientific reporting with however limitations regarding the population size as this data only applies to individuals who have access to health insurance. An explanation of this ,major limitation must be given.

===

We look forward to receiving your revised manuscript.

Kind regards,

Preenan Pillay

Academic Editor

PLOS ONE

Journal Requirements:

“This work was supported by JSPS KAKENHI (grant number JP 21K10938).”

4. For studies involving third-party data, we encourage authors to share any data specific to their analyses that they can legally distribute. PLOS recognizes, however, that authors may be using third-party data they do not have the rights to share. When third-party data cannot be publicly shared, authors must provide all information necessary for interested researchers to apply to gain access to the data. (https://journals.plos.org/plosone/s/data-availability#loc-acceptable-data-access-restrictions)

a) A description of the data set and the third-party source

b) If applicable, verification of permission to use the data set

c) Confirmation of whether the authors received any special privileges in accessing the data that other researchers would not have

d) All necessary contact information others would need to apply to gain access to the data

5. Please upload a copy of Supporting Information Table 1 which you refer to in your text on page 31.

Additional Editor Comments (if provided):

Thank you for submitting your manuscript to plos one. Please note that the following corrections together with the Reviewers recommendations needs to be incorporated into the manuscript.

1. The statistical analysis applied must be substantiated by supplying the data distribution trends and appropriate justification for the statistical tests applied.

2. The discussion lacks critical analysis and this must be corrected so that the novelty of the research incomparison to work already done can be justified. There are a number of studies done on the prevalence of HDPs globally. This can be used in the critical discussion, and how does your manuscript address the gaps that previous research has over looked. In addition discuss how claims data can be beneficial to scientific reporting with however limitations regarding the population size as this data only applies to individuals who have access to health insurance. An explanation of this ,major limitation must be given.

Reviewers' comments:

Reviewer's Responses to Questions

**Comments to the Author**

1. Is the manuscript technically sound, and do the data support the conclusions?

Reviewer #1: Yes

2. Has the statistical analysis been performed appropriately and rigorously? 

Reviewer #1: Yes

3. Have the authors made all data underlying the findings in their manuscript fully available?

Reviewer #1: No

4. Is the manuscript presented in an intelligible fashion and written in standard English?

Reviewer #1: Yes

5. Review Comments to the Author

Reviewer #1: Thank you for an enjoyable and well-written article.

1. Technical soundness and conclusions supported by the results

Conclusions are fair based on the data presented – as the overall incidence has been increasing over time, the authors surmise that other factors, other than age may also be contributing to the trend (smoking and artificial reproductive technologies are mentioned in the discussion but not speculated about in the conclusion). Conclusion matches the objectives of the study.

Corrections have been made for the challenges of the claims data extraction and the potential for duplication of claims cases (based both on ID numbers and on multiple pregnancies). Adjustment / methodologies have been included and described for both of these challenges. The statistics are largely descriptive with an element of testing for differences of incidences between the different age bands as defined – this has also been handled appropriately. Results presented are those from the defined statistical tests described in the methodology.

2. Have statistical analyses been performed appropriately and vigorously?

Corrections have been made for the challenges of the claims data extraction and the potential for duplication of claims cases (based both on ID numbers and on multiple pregnancies). Adjustment / methodologies have been included and described for both of these challenges. The statistics are largely descriptive with an element of testing for differences of incidences between the different age bands as defined – this has also been handled appropriately. Results presented are those from the defined statistical tests described in the methodology.

3. Have the authors made all underlying data for their findings fully available.

No - the limitations on the data and availability thereof have been explained as restricted in terms of the rules of the Ministry of Health, Labour and Welfare and their requirements for data use. Access to the raw data is restricted and would require approval from the Ministry.

4. Is the manuscript presented in an intelligible fashion and written in standard English?

Yes - the manuscript reads very well and is explained well and in simple English where possible.

Areas for correction and improvement:

Line 187 – the mentioned supplementary table 1 is missing and information not presented. This will be necessary Necessary for review to confirm the estimates on incidence reported.

Line 121 – the Japanese standardised disease codes are not known to all – it may be a good idea to explain these a little better given there are multiple codes applying to the same ICD10 and disease description.

Line 318 – “is older than older gestational age” – should this read “other” or “a factor other than” gestational age?

6. PLOS authors have the option to publish the peer review history of their article (what does this mean?). If published, this will include your full peer review and any attached files.

Reviewer #1: No

---

## [Author Response · Author response to Decision Letter 0]

13 Jan 2025

The following is attached separately as a ‘Response to reviewers_R1’ file.

----------

To Additional Editor Comments:

Thank you so much for reviewing and providing useful comments on our manuscript entitled " Maternal Age and the Rising Incidence of Hypertensive Disorders of Pregnancy: A Comprehensive Analysis of National Claims Data from Japan " (Submission ID: PONE-D-24-52545).

The whole manuscript has been revised and rewritten in accordance with your comments.

Our point-by-point responses are presented below.

1. The statistical analysis applied must be substantiated by supplying the data distribution trends and appropriate justification for the statistical tests applied.

We appreciate your constructive suggestion. We would like to respond to your comments as follows:

In this study, we aimed to evaluate differences in HDP incidence rates across age groups using robust statistical methods tailored for proportion data. The Wilson score method was employed to calculate 95% confidence intervals for HDP incidence rates, as it is well-suited for proportion data, particularly when sample sizes are intermediate-to-large or proportions are close to 0 or 1. To ensure the validity of the analysis, we assumed that HDP incidence follows a binomial distribution, where the number of successes was defined as HDP cases and the number of trials as births within each age group, with each birth modeled as an independent trial with a constant probability of HDP occurrence. The average number of births from 2011 to 2019 was used for each age group to account for annual fluctuations in birth numbers, providing stable and reliable incidence rate estimates. Differences across age groups were first assessed using the Kruskal-Wallis test to detect overall differences in HDP incidence, a non-parametric method that does not assume specific data distributions. Following this, pairwise comparisons between the reference group (25–29 years) and other age groups were conducted using Z-tests for proportions. The 25–29 age group was chosen as the reference because it had the largest number of births (n = 89,337), making it statistically stable and minimizing sampling bias. To address potential Type I errors due to multiple comparisons, Holm’s method was applied to adjust p-values, ensuring statistical validity. The results of these analyses, including HDP incidence rates and their 95% confidence intervals for each age group, are illustrated in Figure 4. These data distribution trends confirmed that the conditions for applying Z-tests were met, as the number of successes (np np) and failures (n (1−p) n (1−p)) in each group satisfied the criterion np ≥ 5 np ≥ 5 and n (1−p) ≥ 5n (1−p) ≥ 5. These rigorous methods ensured the reliability and objectivity of our statistical analysis.

2. The discussion lacks critical analysis and this must be corrected so that the novelty of the research incomparison to work already done can be justified. There are a number of studies done on the prevalence of HDPs globally. This can be used in the critical discussion, and how does your manuscript address the gaps that previous research has over looked. In addition discuss how claims data can be beneficial to scientific reporting with however limitations regarding the population size as this data only applies to individuals who have access to health insurance. An explanation of this ,major limitation must be given.

We appreciate your constructive suggestion. In order to reflect your comments, we have examined the draft content and added the following text to the Strengths and Limitations Part (page 20, lines 361- page 22, lines 387).

This study addresses a significant gap in existing research by evaluating the incidence of hypertensive disorders of pregnancy (HDP) among younger populations, a group that has been largely overlooked in previous studies. Existing research has predominantly focused on older pregnant women or general population-based estimates, often lacking detailed age-stratified analyses. For instance, international studies such as the Global Burden of Disease project have provided valuable insights into overall prevalence trends but have not adequately captured rare conditions like HDP in adolescents. In contrast, Japan’s National Database of Health Insurance Claims and Specific Health Checkups (NDB) allows for comprehensive analyses across a broad range of age groups, including younger individuals, enabling the detailed identification of rare cases and age-specific trends. Leveraging this capability, the present study tracked HDP incidence rates in younger populations over a nine-year period, addressing a critical gap in public health research.

Moreover, one of the strengths of the NDB lies in its foundation on Japan’s universal health insurance system, which covers nearly the entire population. This minimizes selection bias while ensuring robust statistical power. The NDB has been widely recognized as a valuable resource for understanding healthcare utilization trends and disease incidence [28–30] [59, 60]. By utilizing real-world clinical data, it provides critical insights into disease management and patient behavior, complementing findings from prospective cohort studies. This study harnessed these features of the NDB to calculate HDP incidence rates in the Hokkaido region, offering a unique perspective on the epidemiology of HDP.

Another notable strength of this study is its ability to accurately track the annual trends of HDP over a long period, from 2011 to 2019, prior to the onset of the COVID-19 pandemic. Importantly, the study also includes data covering pregnancies among women aged 19 years or younger, an age group that has been challenging to analyze in prior prospective cohort studies conducted in developed countries. This comprehensive inclusion of younger age groups highlights the distinct contribution of this study to understanding the epidemiology of HDP across diverse demographic categories.

However, our study has several limitations. First, ・・・・ 

To Reviewer 1:

Thank you so much for reviewing and providing useful comments on our manuscript entitled " Maternal Age and the Rising Incidence of Hypertensive Disorders of Pregnancy: A Comprehensive Analysis of National Claims Data from Japan " (Submission ID: PONE-D-24-52545).

The whole manuscript has been revised and rewritten in accordance with your comments.

Our point-by-point responses are presented below.

15. Are details of the methodology sufficient to allow the experiments to be reproduced?

Yes – I think the only difficulty may be the reproduction of the same methods in combining ID1 and ID2 numbers from the claims database into a single ID number.

→ We agree with your comment. The description of the calculation method for IDT has been strengthened as follows (page 7, lines 113-120).

Even if the ID1 is different, if the ID2 is the same, it is considered to be the same patient, and even if the ID2 is different, if the ID1 is the same, it is considered to be the same patient (or the ID is reassigned so that it can be considered to be the same).

In practice, the following procedure is used.

STEP 0. All data rows have the ID1, ID2 and IDT columns. Initially, IDT is empty.

STEP 1. Assign the same IDT to the same ID1.

STEP 2. Make sure that the same IDT is used for the same ID2.

STEP 3. Make sure that the same IDT is used for the same ID1.

STEP 4. Repeat steps 2 and 3 until the number of updates reaches zero. 

Areas for correction and improvement:

• Line 187 – the mentioned supplementary table 1 is missing and information not presented. Necessary for review to confirm the estimates.

We apologize for the lack of attachments in the first draft we submitted. We have added supplementary table 1, which was missing. We would appreciate your confirmation.

• Line 121 – the Japanese standardised disease codes are not known to all – it may be a good idea to explain these a little better given there are multiple codes applying to the same ICD10 and disease description.

We appreciate your constructive suggestion. We have added the following text to the explanation for Table 1 (page 9, lines 138-150).

‘The Japanese standardized disease codes are part of a comprehensive classification system used in Japan to identify and record diseases and conditions. These codes are linked to the International Classification of Diseases, 10th Revision (ICD-10), but include additional details to accommodate Japan's healthcare system and billing practices. While some ICD-10 codes directly map to a single Japanese standardized disease code, others may correspond to multiple codes depending on the context and specific details recorded in medical receipts. For instance, codes for gestational hypertension and preeclampsia may vary based on the level of detail provided in the diagnosis or treatment description. To ensure consistency, this study reviewed and standardized 15 Japanese disease codes explicitly associated with hypertensive disorders in pregnancy (HDP). The mapping process involved careful evaluation of the disease descriptions and their correspondence to ICD-10 codes, as summarized in Table 1. This approach ensures that the selected codes accurately represent HDP cases within the constraints of the NDB.’

• Line 318 – “is older than older gestational age” – should this read “other” or “a factor other than” gestational age?

We apologize for the typographical error in the first submitted draft. We have corrected it and made the aim of the logistic regression analysis clear as follows.

‘is older than older gestational age’

→ “is a factor other than gestational age” (page 19, lines 342).

---

## [Editor Report · Decision Letter 1]

29 Jan 2025

Maternal Age and the Rising Incidence of Hypertensive Disorders of Pregnancy: A Comprehensive Analysis of National Claims Data from Japan.

PONE-D-24-52545R1

Dear Dr. Koyama,

We’re pleased to inform you that your manuscript has been judged scientifically suitable for publication and will be formally accepted for publication once it meets all outstanding technical requirements.

Kind regards,

Preenan Pillay

Academic Editor

PLOS ONE

Additional Editor Comments (optional):

I would like to thank the Authors for their revision to the manuscript and have accepted it based on the revisions made.

It is however recommended that the manuscript is re checked for copy-editing and language.

---

## [Editor Report · Acceptance letter]

PONE-D-24-52545R1

PLOS ONE

Dear Dr. Koyama,

I'm pleased to inform you that your manuscript has been deemed suitable for publication in PLOS ONE. Congratulations! Your manuscript is now being handed over to our production team.

Kind regards,

on behalf of

Prof Preenan Pillay

Academic Editor

PLOS ONE